# Electronic patient-reported outcome measures using mobile health technology in rheumatology: A scoping review

**Jaclyn Shelton**[1], **Sierra Casey**[1], **Nathan Puhl**[1], **Jeanette Buckingham**[1], **Elaine Yacyshyn**[2]*

**1** Faculty of Medicine and Dentistry, University of Alberta, Edmonton, Alberta, Canada, **2** Department of Medicine, University of Alberta, Edmonton, Alberta, Canada

* eyacyshyn@ualberta.ca

## Abstract

### Objective

This scoping review aims to characterize the current literature on electronic patient-reported outcome measures (ePROMs) in rheumatology and assess the feasibility and utility of ePROMs and mobile health technology in the management of rheumatic disease.

### Introduction

Patient-reported outcome measures (PROMs) are commonly used in rheumatology as they are important markers of disease activity and overall function, encourage shared decision-making, and are associated with high rates of patient satisfaction. With the widespread use of mobile devices, there is increasing interest in the use of mobile health technology to collect electronic PROMs (ePROM).

### Inclusion criteria

All primary studies that involve the collection of ePROMs using mobile devices by individuals with a rheumatic disease were included. Articles were excluded if ePROMs were measured during clinic appointments.

### Methods

A scoping review was performed using Medline, Embase, PsycINFO, and CINAHL with index terms and key words related to "patient-reported outcome measures", "rheumatic diseases", and "mobile health technology".

### Results

A total of 462 records were identified after duplicates were removed. Of the 70 studies selected for review, 43% were conference proceedings and 57% were journal articles, with the majority published in 2016 or later. Inflammatory arthritis was the most common rheumatic disease studied. Generic ePROMs were used over three times more often than disease-specific ePROMs. A total of 39 (56%) studies directly evaluated the feasibility of

**Data Availability Statement:** All relevant data are within the paper and its Supporting information files.

**Funding:** The author(s) received no specific funding for this work.

**Competing interests:** The authors have declared that no competing interests exist.

ePROMs in clinical practice, 19 (27%) were clinical trials that used ePROMs as study endpoints, 9 (13%) were focus groups or surveys on smartphone application development, and 3 (4%) did not fit into one defined category.

## Conclusion

The use of ePROMs in rheumatology is a growing area of research and shows significant utility in clinical practice, particularly in inflammatory arthritis. Further research is needed to better characterize the feasibility of ePROMs in rheumatology and their impact on patient outcomes.

## Introduction

Rheumatic diseases are among the most common chronic diseases worldwide, with recent United States data citing a prevalence rate of 22.7% from 2013 to 2015 [1]. Rheumatic diseases are a leading cause of disability and are associated with a loss of productivity, reduced quality of life, and place a significant burden on the health care system [2]. Fortunately, long-term outcomes of rheumatic diseases have improved remarkably in recent years as a result of new pharmacologic treatments, improved disease management strategies, and enhanced outcome assessments for follow-up [3].

Patient-reported outcomes (PROs) are measures that allow patients to effectively communicate their own experience of their disease to their health care providers [4] and have become increasingly utilized in the field of rheumatology [5–7]. Patient-reported outcome measures (PROM) are quantifiable tools used to measure PROs [8]. PROMs have been used in many rheumatic diseases, including rheumatoid arthritis (RA), ankylosing spondylitis (AS), psoriatic arthritis (PsA), gout, osteoarthritis (OA), systemic lupus erythematosus (SLE), fibromyalgia, osteoporosis, juvenile idiopathic arthritis (JIA), and myositis [4, 9–12].

PROMs are used in clinical practice for a variety of purposes. They may be disease-specific or may evaluate general aspects of health [8]. In rheumatology, they are most often used as screening tools to characterize physical, social, and emotional functioning that is generally not detected in regular clinic visits [13]. They have been shown to encourage shared decision-making and are associated with high rates of patient satisfaction [14]. PROMs are also used to monitor disease activity over time and have been integral in the treat-to-target (T2T) approach used in the management of many inflammatory rheumatic diseases including RA, PsA, SLE, and AS [15–17]. The T2T approach involves the frequent assessment of disease activity and adjustment of disease modifying anti-rheumatic drugs (DMARDs) in order to induce remission or achieve a low disease activity state [3]. Current guidelines on the treatment of newly diagnosed or active RA states patients should be reassessed every 1 to 3 months to ensure that treatment targets are being met [3, 18]. This tight disease control can prevent complications such as joint destruction and long-term disability [19, 20]. However, as a result of limited resources, such frequent assessments are not always feasible. Further, assessments at appointments alone dismiss fluctuations of disease activity between appointments and recall bias can influence a patient's report of disease flares [21].

With the widespread use of mobile devices such as mobile phones, smartphones, smartwatches, and tablets, there is increasing interest in the use of these devices to collect electronic PROMs (ePROM) outside of clinic appointments [22]. A number of reviews have evaluated the feasibility of ePROMs and mobile health technologies in single rheumatic diseases,

including OA [23] and SLE [24]. However, there are no known reviews that have characterized the landscape of ePROMs in the field of rheumatology as a whole, which limits our ability to critically evaluate the feasibility and utility of ePROMs in general rheumatology practice. In this scoping review, we aim to characterize the current literature on ePROMs in rheumatology and assess the feasibility and utility of ePROMs and mobile health technology in the management of rheumatic disease.

## Review questions

The following questions were used to guide our scoping review:

1. How are ePROMs collected in rheumatology?

2. Which ePROMs are measured using mobile health technology in rheumatology?

3. Is the collection of ePROMs using mobile health technology in rheumatology feasible?

## Inclusion criteria

### Participants

This review considered studies that included individuals of all ages with a rheumatic disease. Rheumatic diseases included RA, PsA, OA, spondyloarthropathy, gout, SLE, fibromyalgia, osteoporosis, JIA, myositis, and vasculitis.

### Concept

The primary concept explored in this review was the collection of ePROMs using a mobile device in rheumatology. All generic and disease-specific ePROMs were included. Studies that did not specify which ePROM was used were excluded. Mobile devices included mobile phones, smartphones, smartwatches, and tablets. ePROMs accessed through computers were excluded. Passively-collected patient-data such as measurements from accelerometers as well as subjective and non-quantifiable patient-reported data such as symptom diaries were excluded.

### Context

Studies completed in all geographic locations were selected for review. Studies were included if ePROMs were collected between rheumatology appointments and were excluded if ePROMs were completed in clinic waiting rooms or during clinic appointments.

### Evidence sources

All primary studies, including conference proceedings and journal articles, as well as dissertations/theses were considered for review. Systematic reviews and meta-analyses were excluded. Duplicate conference abstracts presented at more than one conference and conference abstracts subsequently published as a journal article were excluded. Studies including preliminary results of a later published study were also excluded.

## Methods

The objectives, inclusion criteria, and methods of this scoping review were developed in advance using the Joanna Briggs Institute (JBI) Reviewer's Manual on Scoping Reviews [25].

## Search strategy

The search strategy was developed in collaboration with research librarians at the John Scott Library at the University of Alberta in Edmonton, Alberta, Canada. The search strategy comprised of index terms and key words related to "rheumatic diseases", "mobile health technology", and "patient-reported outcome measures", which included a validated search strategy for PROMs developed by the Patient-Reported Outcome Measurement Group of the University of Oxford [26]. The search was conducted using Ovid Medline, Ovid Embase, Ovid PsycINFO, and EBSCO CINAHL and the index terms were translated between databases. The detailed search strategy is listed in S1 Appendix. The search was completed on March 8, 2020 and included all articles published prior to this date with no restrictions in regard to publication language. All existing literature extracted from the databases was reviewed, including conference proceedings, journal articles, and dissertation/theses.

## Source of evidence screening and selection

The title and abstract screening was completed by two reviewers (JS, SC) independently using pre-established inclusion criteria. Full-texts were then reviewed independently by the same two reviewers (JS, SC). All included conference abstracts were then reviewed to look for corresponding journal articles that were not captured by our search strategy. Disagreements between reviewers at all stages of study screening and selection were resolved through discussion and a third reviewer (EY) was involved if a consensus could not be reached.

## Data extraction

All references were entered into RefWorks and subsequently extracted into an excel worksheet that cited the title, author(s), and year of publication. Relevant data was extracted by two independent reviewers (JS, SC) using an adapted JBI instrument for data extraction [25]. The following information was extracted from each study: study objective(s), population characteristics (including the number of participants, mean age, and type of rheumatic disease), type of ePROM collected, frequency of ePROM collection, length of study, type of electronic device used for ePROM collection, geographic location of the study, and main findings of the study.

# Results

## Search results

A total of 462 records were identified after duplicates were removed. Titles and abstracts were then screened for suitability, followed by full-text reviews. Included conference abstracts were reviewed and a total of eight corresponding journal articles, not captured by our original search criteria, were added to the analysis. In addition, two journal articles cited in a review article, captured by our search criteria, were added for review. In total, 70 journal articles and conference proceedings were included in the final data analysis (S2 Appendix). The inclusion process is outlined in Fig 1 using an adapted PRISMA flowchart [27].

## Study characteristics

Of the studies included in our review, 43% were conference proceedings (N = 30), while 57% were journal articles (N = 40). Fifty-four (77%) studies were published in 2016 or later (Fig 2). The majority of the studies were from Europe (N = 25), North America (N = 25), and Asia (N = 15), with fewer from Oceania (N = 4) and South America (N = 1). The greatest number of studies came from the United States (N = 24), the United Kingdom (N = 9), and China

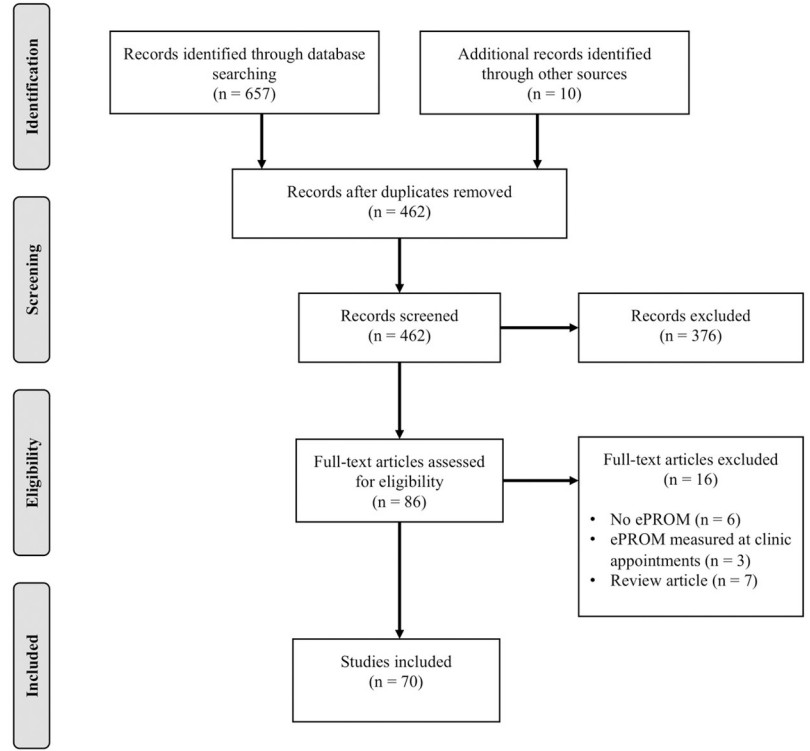

**Fig 1. PRISMA flow diagram of the review process.**

(N = 11). Of the studies that specified study duration (N = 43), 21 (49%) were less than or equal to 3 months in duration. Population sizes were generally small as 47 (67%) studies had a population size of less than or equal to 100 participants.

## Patient characteristics

Study participants included both adults and children with rheumatic diseases. A total of 83% of the studies in our review included adult participants (N = 58) and 17% included children ≦18 years old (N = 12). Of the adult populations studied, the most common mean age range was between 50 and 60 years old. Inflammatory arthritis, including RA and PsA, was the most common rheumatic disease studied (N = 34) (Fig 3). Eleven studies evaluated patients with JIA and eight evaluated patients with OA. Other rheumatic diseases studied included SLE (N = 3), fibromyalgia (N = 3), spondyloarthropathy (N = 3), and gout (N = 1).

## ePROM characteristics

A total of 67% of studies in our review used smartphones to collect ePROMs (N = 47). Seven studies used multiple smart devices, eight studies used handheld computers or tablets, five studies used mobile phone text-messages, and three studies used smartwatches. There were 54 different ePROMs identified in our review (Tables 1 and 2). Generic ePROMs were used 174 different times throughout the 70 studies included in this review (Table 1). By contrast, disease-specific ePROMs were used 46 different times. Of the disease-specific ePROMs, the Health Assessment Questionnaire (HAQ) was the most commonly measured (Table 2).

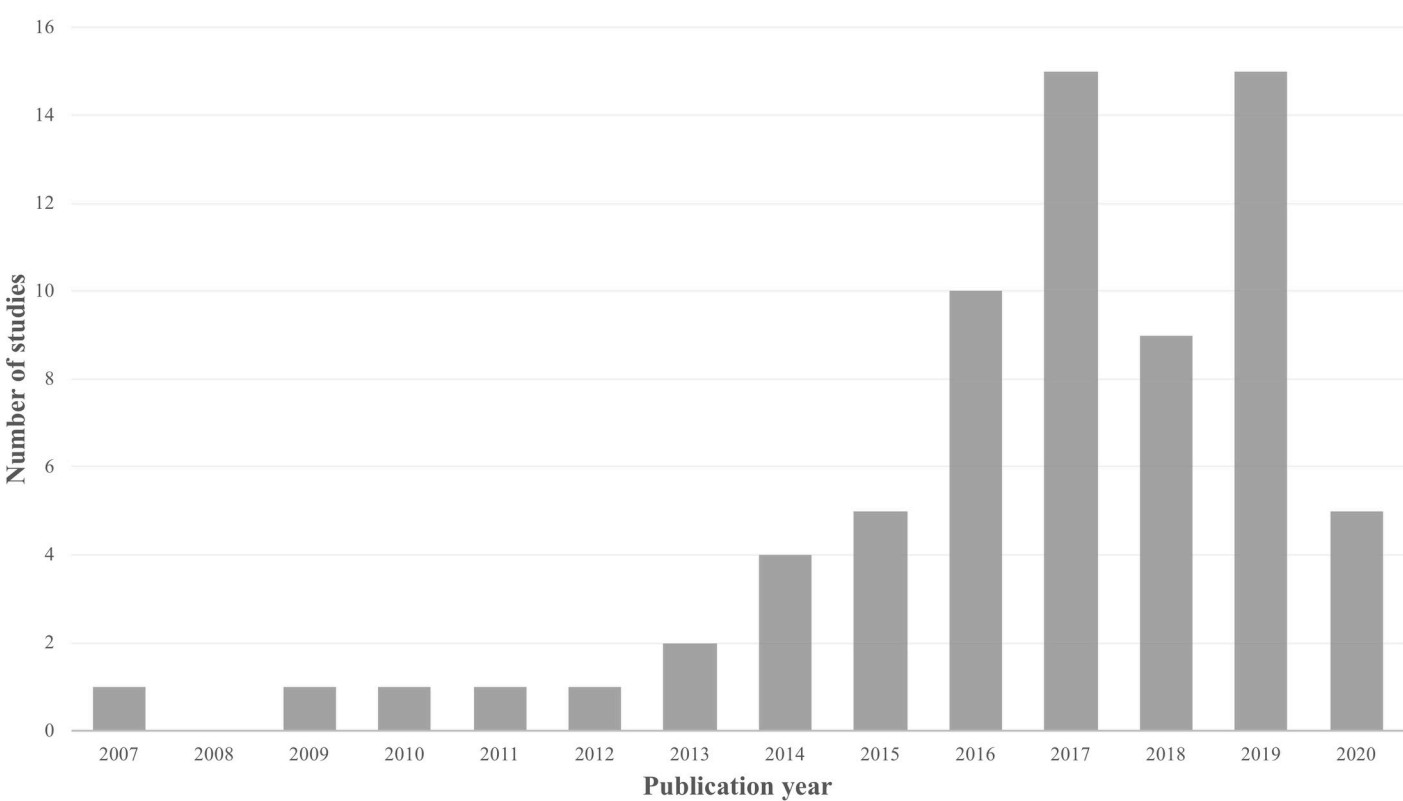

**Fig 2. Year of publication.**

## Purpose of studies

A total of 56% of studies in our review directly evaluated ePROMs in clinical practice through pilot studies, cohort studies, or randomized controlled trials (N = 39). Nineteen studies (27%) were clinical trials that used ePROMs as study endpoints. Nine studies (13%) were reports on smartphone application development using focus groups or surveys. Three studies (4%) had multiple purposes or did not fit into one defined category.

## Discussion

This scoping review provides an overview of the literature surrounding the collection of ePROMs using mobile heath technology in rheumatology. This is a growing area of research, with 77% of the 70 included studies published in the past 5 years. The majority of studies focused on adults with inflammatory arthritis. ePROMs were most commonly measured using smartphones and generic ePROMs were used over three times more often than disease-specific ePROMs. The majority of studies evaluated ePROMs in clinical settings, which suggests there is increasing significance and utility of ePROMs in clinical practice today.

The majority of studies analyzed in our review were published in the past five years. The growth of research on this topic in recent years is likely a result of the widespread adoption of smart technology. As of 2019, 81% of adults in the United States owned a smartphone, which is an increase of 35% since 2011 [28]. The use of PROMs in daily rheumatology practice has also increased over the past decade. A study by Wolfe and Pincus noted that in 1998, just 15% of rheumatologists utilized quantifiable PROMs concerning health status measures [29].

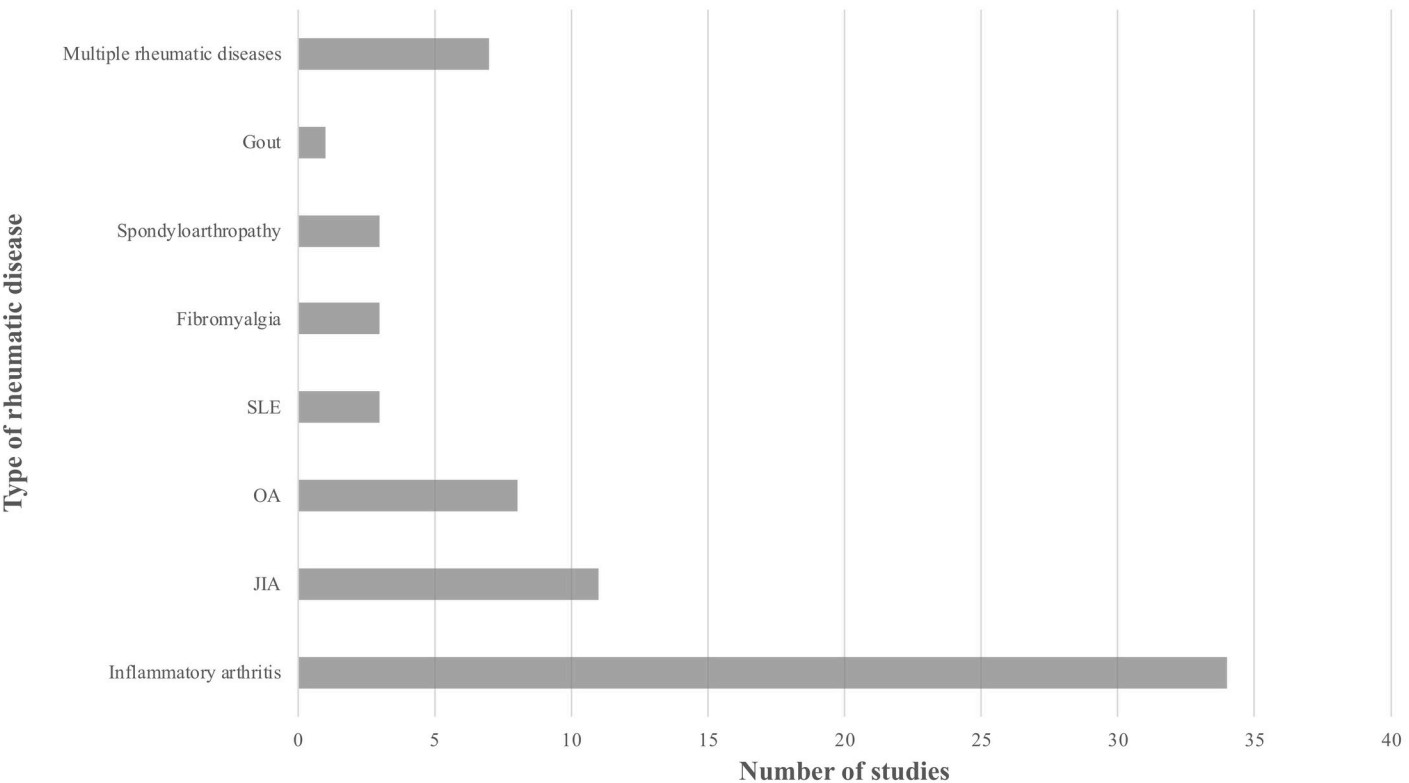

**Fig 3. Types of rheumatic disease studied.**

Today, PROMs are central in the field of rheumatology and organizations such as the American College of Rheumatology and the European League Against Rheumatism recommend the use of health-related quality of life and general well-being PROMs as a means to determine whether treatment targets are being met [5].

The majority of studies in our review were focused on RA. This is likely due to its greater prevalence when compared to other inflammatory rheumatic diseases, as recent data suggests it affects as many as 1.36 million adults in the United States [30]. JIA was the second most common disease studied and this may be influenced by the enhanced proficiency and widespread use of smart technology amongst younger adults and children. A 2018 Statistics Canada survey showed that 98% of internet users aged 15 to 24 own a cellphone, when compared to 87% of adults aged 45 to 64 [31]. Additionally, young people aged 15 to 24 are much more likely to check their smart devices at least every 30 minutes (57.5% versus 39.5%) [31]. This increased engagement with smart devices may explain why ePROMs are so frequently studied in the context of JIA.

The majority of ePROMs found in our review were generic, which is consistent with previous literature. A study by Hiligsmann et al. reported that the most common PROMs used in RA were generic, including the SF-36 and EQ-5D [32]. This suggests that the types of PROMs used in rheumatology have remained stable over the past several years and have not varied appreciably across different PROM delivery platforms.

Several studies included in our review evaluated the correlation of ePROMs with physician-assessed disease activity. For example, a study by Nishigushi et al. described a 3-month trial involving a smartphone application that collected ePROMs in patients with RA [33]. Their

**Table 1. Generic ePROMs.**

| Type of ePROM | Constructs assessed | ePROM | Number of times reported |
|---|---|---|---|
| Dimension-specific | Signs & symptoms | GSQ | 1 |
| | | Pain | 33 |
| | | Stiffness | 13 |
| | | Swelling | 2 |
| | | SJC/TJC | 8 |
| | | Sleep | 11 |
| | | General fatigue | 20 |
| | | FACIT-F | 3 |
| | | Flares | 2 |
| | | Fever | 1 |
| | | Rash | 1 |
| | Global judgements | PGA | 4 |
| | | General health/wellbeing | 5 |
| | | Disease activity/severity | 3 |
| | Physical function | FFbH | 2 |
| | | Function | 8 |
| | | Physical activity | 4 |
| | Psychological well-being | General mood | 12 |
| | | HADS | 3 |
| | | Emotional distress | 1 |
| | | Sense of relaxation | 1 |
| | | Coping | 1 |
| | | PHQ-2 | 1 |
| | | Stress/negative events | 1 |
| | Social well-being | Time spent outside | 1 |
| | | Pain interference | 8 |
| | | Participation in social activities | 1 |
| | Role activities | Capacity to work/impact on work | 2 |
| | | Capacity to complete daily activities | 3 |
| | | Employment status | 1 |
| | | Sick leave | 3 |
| | | Hours worked | 1 |
| | Self-management | PAM | 1 |
| | | P-SEMS | 1 |
| | Quality of life | Quality of life | 1 |
| | | SF-36 | 3 |
| | | EQ-5D | 2 |
| | Other | Medication use | 1 |
| | | Adverse events | 2 |
| | | Drug compliance | 1 |
| | | Clinic visits | 1 |
| | | **Total** | **174** |

GSQ, General Symptom Questionnaire; SJC, swollen joint count; TJC, tender joint count; FACIT-F, Functional Assessment of Chronic Illness Therapy—Fatigue; PGA, Patient Global Assessment; FFbH, Hannover Functional Ability Questionnaire; HADS, Hospital Anxiety and Depression Scale; PHQ-2, Patient Health Questionnaire-2; PAM, Patient Activation Measure; P-SEMS, Patient-Reported Outcomes Measurement Information System Self-Efficacy in Managing Symptoms; SF-36, 36-Item Short Form Health Survey; EQ-5D, EuroQol-5D.

**Table 2. Disease-specific ePROMs.**

| Type of ePROM | Rheumatic disease | ePROM | Number of times reported |
|---|---|---|---|
| Disease-specific | Inflammatory arthritis | RAPID3 & RAPID4 | 9 |
| | | RADAI | 4 |
| | | DAS28 | 7 |
| | | HAQ | 13 |
| | | RAID | 1 |
| | | RA-WIS | 1 |
| | Spondyloarthropathy | BASDAI | 2 |
| | | ASDAS | 1 |
| | | BASFI | 1 |
| | SLE | SLEDAI | 3 |
| | OA | WOMAC NRS | 2 |
| | | KOO | 1 |
| | Fibromyalgia | FIQR | 1 |
| | | **Total** | 46 |

RAPID3, Routine Assessment of Patient Data 3; RAPID4, Routine Assessment of Patient Data 4; RADAI, Rheumatoid Arthritis Disease Activity Index; DAS28, Disease Activity Score; HAQ, Health Assessment Questionnaire; RAID, Rheumatoid Arthritis Impact of Disease; RA-WIS, Rheumatoid Arthritis Work Instability Scale; BASDAI, Bath Ankylosing Spondylitis Disease Activity Index; ASDAS, Ankylosing Spondylitis Disease Activity Score; BASFI, Bath Ankylosing Spondylitis Functional Index; SLEDAI, Systemic Lupus Erythematous Disease Activity Index; WOMAC NRS, Western Ontario and McMaster Universities Osteoarthritis Index Numeric Rating Scale; KOOS, Knee Injury and Osteoarthritis Outcome Score; FIQR, Revised Fibromyalgia Impact Questionnaire.

application involved a DAS28 predictive model of disease activity, which included patient measurements of tender joint counts, a modified heath assessment questionnaire (mHAQ), as well as objective gait balance measurements [33]. They found that the DAS28 predictive model correlated with DAS28 scores completed by physicians, which suggests that their application could accurately assess disease activity over time [33]. The application was also viewed favourably as participants noted repeated objective assessments through ePROMs were helpful in managing their disease [33].

Several studies included in our review argued that the use of ePROMs in rheumatology have led to favourable patient outcomes [34]. A study by Huang et al. looked at the outcomes of SLE patients after completing disease-specific ePROMs over time [34]. Of the 1,090 patients enrolled in their study, they found that patients who completed a greater number of self-assessments had a lower probability of relapse and a higher likelihood of achieving remission [34]. This suggests that frequent reporting of ePROMs can lead to improved disease management. Unfortunately, the benefits associated with frequent ePROMs can be limited by poor patient compliance and high attrition rates. A study by Beukenhorst et al. looked at a smartwatch application to collect ePROMs in patients with OA over a 3-month period, and noted the number of participants decreased from 26 to 11 participants at the end of 3 months [35]. Moreover, a study by Nowell et al. looked at patient engagement within a large patient registry, ArthritisPower, and found 20.6% of patients never completed a single ePROM [36].

Our study has several limitations. The inclusion criteria in our study was broad given the all-encompassing nature of a scoping review. This resulted in significant heterogeneity between our studies which limited our ability to compare their findings. A future systematic review and meta-analysis should assess the real-world use and feasibility of using ePROMs in rheumatology practice and assess the ideal frequency of ePROM collection to maximize patient engagement. In addition, larger studies over greater periods of time should be

conducted to demonstrate whether the use of ePROMs are not only feasible but improve patient outcomes and disease activity over time.

## Conclusion

This scoping review highlights the current literature on the collection of ePROMs using mobile health technology in rheumatology. The use of ePROMs in rheumatology is a growing area of research, most notably in inflammatory arthritis. The majority of studies evaluated ePROMs in clinical settings, which suggests there is increasing adoption and utility of ePROMs in clinical practice today. Further high-quality research is needed to better characterize the feasibility of ePROMs in rheumatology and its impact on patient outcomes.

## Supporting information

**S1 Appendix. Search strategy (Ovid Medline).**
(PDF)

**S2 Appendix. Studies included in the scoping review.**
(PDF)

**S3 Appendix. PRISMA-ScR checklist.**
(PDF)

## Author Contributions

**Conceptualization:** Jaclyn Shelton, Nathan Puhl, Elaine Yacyshyn.

**Data curation:** Jaclyn Shelton, Sierra Casey, Elaine Yacyshyn.

**Formal analysis:** Jaclyn Shelton, Sierra Casey, Elaine Yacyshyn.

**Investigation:** Jaclyn Shelton, Elaine Yacyshyn.

**Methodology:** Jaclyn Shelton, Jeanette Buckingham, Elaine Yacyshyn.

**Project administration:** Jaclyn Shelton, Elaine Yacyshyn.

**Supervision:** Elaine Yacyshyn.

**Writing – original draft:** Jaclyn Shelton, Sierra Casey, Elaine Yacyshyn.

**Writing – review & editing:** Jaclyn Shelton, Sierra Casey, Jeanette Buckingham, Elaine Yacyshyn.

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
