## [Decision Letter · Decision Letter 0]

29 Mar 2021

PONE-D-20-36935

Electronic patient-reported outcome measures using mobile health technology in rheumatology: A scoping review

PLOS ONE

Dear Dr. Yacyshyn,

Thank you for submitting your manuscript to PLOS ONE. After careful consideration, we feel that it has merit but does not fully meet PLOS ONE’s publication criteria as it currently stands. Therefore, we invite you to submit a revised version of the manuscript that addresses the points raised during the review process.

ACADEMIC EDITOR

Please revise the manuscript based on the reviewers' comments and re-submit to PLOS ONE.

We look forward to receiving your revised manuscript.

Kind regards,

Md Asiful Islam, Ph.D.

Academic Editor

PLOS ONE

Journal Requirements:

2. This manuscript reports a scoping review; while PLOS ONE does not consider narrative reviews, we do consider systematic and scoping reviews, in which authors address a clearly defined research question; conduct a systematic and comprehensive literature review; and use clearly reported, reproducible, and systematic methods to identify, select, and extract data from relevant research. A scoping review (also scoping study) refers to a rapid gathering of literature in a given policy or clinical area where the aims are to accumulate as much evidence as possible and map the results. Scoping reviews are a type of literature review that aims to provide an overview of the type, extent and quantity of research available on a given topic. By ‘mapping’ existing research, a scoping review can identify potential research gaps and future research needs, and do so by using systematic and transparent methods. For guidance in the assessment of this scoping review, you might refer to a newly developed extension of the PRISMA checklist (PRISMA-ScR), which is available at https://www.acpjournals.org/doi/10.7326/M18-0850

Please assess whether the conclusions are supported by the review method and results, note that scoping reviews should not be used to address clinically meaningful questions or attempt to provide implications for clinical practice. We ask that you please invite independent reviewers qualified to assess scoping review methods, as well as reviewers with specific expertise in the subject area. To discuss this work, please contact me.

Additional Editor Comments (if provided):

Reviewers' comments:

Reviewer's Responses to Questions

**Comments to the Author**

1. Is the manuscript technically sound, and do the data support the conclusions?

Reviewer #1: Yes

Reviewer #2: Yes

2. Has the statistical analysis been performed appropriately and rigorously? 

Reviewer #1: Yes

Reviewer #2: Yes

3. Have the authors made all data underlying the findings in their manuscript fully available?

Reviewer #1: Yes

Reviewer #2: Yes

4. Is the manuscript presented in an intelligible fashion and written in standard English?

Reviewer #1: Yes

Reviewer #2: Yes

5. Review Comments to the Author

Reviewer #1: The review by Yacushyn et al. has been conducted according to Joanna Briggs Institute Manual on Scoping Reviews. The methods of the literature search seem valid; however, I was a little surpised to find out that for example our paper on automated monitoring of early RA using SMS messages published online 11 February 2019 was included as a conference abstract (https://doi.org/10.1002/acr.23846) and not as a full paper. Therefore, I suggest that the authors confirm that the conference abstracts in the reference list actually have been published only as abstracts by March 2020. Concerning the methods, I have no further comments – the methods have been described in detail and the search strategy is relevant.

In table 1 two very common quality of life measures, SF-36 and EQ-5D are for some reason listed in the section “other” - a construct of Quality of life should be added – QoL is one of the most central PROMs!

Concerning table 2 I have also a comment: I would not refer to DAS28 or HAQ as a disease specific measures for Rheumatoid arthritis – both are used widely in different types of inflammatory arthritis, like undifferentiated arthritis, psoriatic arthritis and peripheral spondyloarthropathy.

Conclusions are presented clearly. As a general point I would like to comment that certainly ePROMs will be used increasingly in the future instead of papers filled out in the clinic - it remains to be seen how much research will be conducted on this inevitable change in the way of collecting patient reported data (it should be remembered that the number of the commercial health apps is increasing rapidly). For example, in Finland multiple rheumatology clinics use routinely remote monitoring of PROMs like HAQ, pain VAS and fatigue etc. Personally, I think the question remains how often patients are motivated to fill out PROMs and how they adhere to it – would appropriate interval be weekly or quarterly?

Reviewer #2: The authors conducted a systematic review from several databases to characterize the current literature on ePROMs in rheumatology and assess the feasibility and utility of ePROMs and mobile health technology in the management of rheumatic

disease. 76 studies were reviewed.

The abstract should state how many papers the search yielded: "A total of 470 records were identified after duplicates were removed."

I think this is a good study. The language is fluent. I cannot find methodological flaws. I this this manuscript could be published if the number of matches from the search is added to the abstract.

6. PLOS authors have the option to publish the peer review history of their article (what does this mean?). If published, this will include your full peer review and any attached files.

Reviewer #1: No

Reviewer #2: No

---

## [Author Response · Author response to Decision Letter 0]

5 Jun 2021

We thank the Editor, and the reviewers for your review of our paper. 

We have responded to the specific reviewer comments in the uploaded document.

---

## [Editor Report · Decision Letter 1]

9 Jun 2021

Electronic patient-reported outcome measures using mobile health technology in rheumatology: A scoping review

PONE-D-20-36935R1

Dear Dr. Yacyshyn,

We’re pleased to inform you that your manuscript has been judged scientifically suitable for publication and will be formally accepted for publication once it meets all outstanding technical requirements.

Kind regards,

Md Asiful Islam, Ph.D.

Academic Editor

PLOS ONE
---

## [Editor Report · Acceptance letter]

13 Jul 2021

PONE-D-20-36935R1 

Electronic patient-reported outcome measures using mobile health technology in rheumatology: A scoping review 

Dear Dr. Yacyshyn:

I'm pleased to inform you that your manuscript has been deemed suitable for publication in PLOS ONE. Congratulations! Your manuscript is now with our production department. 

Kind regards, 

on behalf of

Dr. Md Asiful Islam 

Academic Editor

PLOS ONE